# Rosmarinic Acid Exhibits a Lipid-Lowering Effect by Modulating the Expression of Reverse Cholesterol Transporters and Lipid Metabolism in High-Fat Diet-Fed Mice

**DOI:** 10.3390/biom11101470

**Published:** 2021-10-06

**Authors:** Jean Baptiste Nyandwi, Young Shin Ko, Hana Jin, Seung Pil Yun, Sang Won Park, Hye Jung Kim

**Affiliations:** 1Department of Pharmacology, Institute of Health Sciences, College of Medicine, Gyeongsang National University, Jinju 52727, Korea; nbaptiste1988@gmail.com (J.B.N.); shini33@naver.com (Y.S.K.); hanajin.kr@daum.net (H.J.); spyun@gnu.ac.kr (S.P.Y.); parksw@gnu.ac.kr (S.W.P.); 2Department of Convergence Medical Science (BK21 Plus), Gyeongsang National University, Jinju 52727, Korea; 3Department of Pharmacy, School of Medicine and Pharmacy, College of Medicine and Health Sciences, University of Rwanda, Kigali 4285, Rwanda

**Keywords:** ABC transporters, AMPK, hyperlipidemia, RCT, rosmarinic acid

## Abstract

Hyperlipidemia is a potent risk factor for the development of cardiovascular diseases. The reverse cholesterol transport (RCT) process has been shown to alleviate hyperlipidemia and protect against cardiovascular diseases. Recently, rosmarinic acid was reported to exhibit lipid-lowering effects. However, the underlying mechanism is still unclear. This study aims to investigate whether rosmarinic acid lowers lipids by modulating the RCT process in high-fat diet (HFD)-induced hyperlipidemic C57BL/6J mice. Our results indicated that rosmarinic acid treatment significantly decreased body weight, blood glucose, and plasma total cholesterol and triglyceride levels in HFD-fed mice. Rosmarinic acid increased the expression levels of cholesterol uptake-associated receptors in liver tissues, including scavenger receptor B type 1 (SR-B1) and low-density lipoprotein receptor (LDL-R). Furthermore, rosmarinic acid treatment notably increased the expression of cholesterol excretion molecules, ATP-binding cassette G5 (ABCG5) and G8 (ABCG8) transporters, and cholesterol 7 alpha-hydroxylase A1 (CYP7A1) as well as markedly reduced cholesterol and triglyceride levels in liver tissues. In addition, rosmarinic acid facilitated fatty acid oxidation through AMP-activated protein kinase (AMPK)-mediated carnitine palmitoyltransferase 1A (CPT1A) induction. In conclusion, rosmarinic acid exhibited a lipid-lowering effect by modulating the expression of RCT-related proteins and lipid metabolism-associated molecules, confirming its potential for the prevention or treatment of hyperlipidemia-derived diseases.

## 1. Introduction

Hyperlipidemia is a well-known risk factor for atherosclerosis and subsequent atherosclerotic cardiovascular diseases that are associated with high morbidity and mortality in obese and diabetic patients worldwide. Clinically obese and diabetic patients present abnormal lipid homeostasis, placing them at high risk of suffering from heart diseases and stroke [1]. Several studies have provided compelling evidence of hyperlipidemia in response to lipid metabolism disorders in the context of high-fat diets [2,3]. An important pool of evidence has demonstrated that lipid metabolism disorder is characterized by the augmentation of plasma triglycerides and total cholesterol (TC) and low-density lipoprotein (LDL) cholesterol levels.

Lipid homeostasis, particularly cholesterol homeostasis, is a complex process that involves cholesterol uptake, synthesis, metabolism, and excretion [4,5]. Indeed, impaired lipid homeostasis leads to the development of hyperlipidemia. The liver serves as a master regulator of cholesterol synthesis and metabolism, cholesterol degradation to bile acids, and ultimate excretion to bile [6,7]. Excess cholesterol in extrahepatic tissues is subjected to the reverse cholesterol transport (RCT) process and transported by high-density lipoprotein (HDL) or LDL molecules back to the liver for metabolism and excretion [8]. Considerable evidence has demonstrated that RCT plays an important role in improving hyperlipidemia and related diseases [9,10]. The initial step of RCT is facilitated by extracellular acceptors, apo-lipoprotein AI (Apo-AI) or plasma HDL, which take up free cholesterol transported by cellular ATP-binding cassette transporter A1 (ABCA1) and G1 (ABCG1), respectively [11].

HDL-containing cholesterol subsequently binds hepatocyte scavenger receptor (SR)-BI, which mediates cholesterol uptake by the liver [12]. Alternatively, non-HDL cholesterol can be delivered to the liver through hepatocyte LDL receptor (LDL-R) [13]. Indeed, the liver ultimately excretes received cholesterol in the form of free cholesterol through ATP-binding cassette G5 (ABCG5) and G8 (ABCG8) transporters [14,15]. Furthermore, liver cholesterol can be transformed into bile acids and finally eliminated through feces [16]. Therefore, enhancing RCT by increasing cholesterol uptake by the liver and subsequently increasing hepatic cholesterol excretion may improve hyperlipidemia.

Considerable lines of evidence have demonstrated that fatty acid β-oxidation plays an important role in mitigating lipid metabolic disorders [17,18]. The overexpression of carnitine palmitoyltransferase 1A (CPT1A), the key enzyme in fatty acid β-oxidation, in high-fat diet-treated and genetically obese mice resulted in decreased liver triglyceride levels [19]. Similarly, the activation of liver AMPK, a well-known regulator of lipid metabolism and a direct upstream regulator of CPT1A, inhibited the accumulation of lipids in mouse liver and the development of nonalcoholic fatty liver disease (NAFLD) [20,21]. Therefore, promotion of fatty acid β-oxidation could be effective in alleviating hyperlipidemia.

Natural products or pharmaceutical products that improve the RCT or fatty acid β-oxidation processes would attenuate excess lipid accumulation in the liver and extrahepatic tissues and protect against hyperlipidemia. Rosmarinic acid exhibits antioxidant, anti-inflammatory, antidiabetic, and cardioprotective activities [22,23,24]. Furthermore, rosmarinic acid inhibits gluconeogenesis and insulin resistance in rats [25]. In our previous in vitro study, we reported that rosmarinic acid effectively reduces oxLDL-induced cholesterol contents under high glucose (HG) conditions in macrophages by enhancing ABCA1 and ABCG1 expression [26]. In this study, we confirmed the protective effects of rosmarinic acid on hyperlipidemia in HFD-fed C57BL/6 mice and investigated the possible mechanism underlying the antihyperlipidemic effect of rosmarinic acid in vivo.

## 2. Materials and Methods

### 2.1. Materials

Rosmarinic acid (536954), metformin hydrochloride (PHR1084), 2-mercaptoethanol (M3148-25ML), protease inhibitor cocktail (P8340-5ML), and antibody against β-actin (a2066) were purchased from Sigma-Aldrich (St. Louis, MO, USA). A mouse IL-1β ELISA kit (ab100704) and antibodies against ABCA1 (ab18180), LDL-R (ab52818), and carnitine palmitoyltransferase 1A (CPT1A; ab53532) were purchased from Abcam (Cambridge, UK). Antibodies against p-AMPKα^Thr172^ (#2531) and AMPK (#2532) were purchased from Cell Signaling Technology (Beverly, MA, USA). An antibody against ABCG5 (bs-5013R) was obtained from Bioss Antibodies (Woburn, MA, USA). Antibodies against ABCG8 (NBP1-71706) and SR-BI (NB400-104) were procured from Novus Biologicals (Littleton, CO, USA). Anti-cholesterol 7 alpha-hydroxylase A1 (CYP7A1) antibody (PA5-100892) was purchased from Invitrogen (Waltham, MA, USA). Accu check (code 222) was obtained from Roche Diagnostics (Mannheim, Germany). A cholesterol fluorometric assay kit (10007640) and triglyceride colorimetric assay kit (10010303) were procured from Cayman Chemical (Ann Arbor, MI, USA).

### 2.2. Experimental Animals and Treatment

Male C57BL/6 mice (16–20 g, 3 weeks old) were purchased from Koatech (Pyeongtaek, Korea), acclimatized for one week, and maintained in the animal facility at Gyeongsang National University (GNU) for the whole period of the experiment. All animal experiments were approved by the Institutional Board of Animal Research at GNU (approval #: GNU-120208-M0004) and performed in accordance with the National Institutes of Health guidelines for laboratory animal care. Mice were housed under alternating 12 h light/dark cycles with a relative humidity of 60 ± 10%, a controlled temperature of 22 ± 2 °C, and free access to food and water. Mice were fed a normal diet (ND, 2018S, 3.1 kcal/g, Harlan Laboratories, Inc., Indianapolis, IN, USA) (*n* = 10) or a high-fat diet (HFD, 60% fat, Research Diets, Inc., New Brunswick, NJ, USA) for 8 weeks (*n* = 40). Food intake and body weight were checked twice a week, and blood glucose concentrations were evaluated monthly in samples obtained from tails using a blood glucose meter. Mice with a fasting blood glucose (FBG) level ≥150 mg/dl were considered hyperglycemic. Once hyperglycemia was achieved, the mice were randomly assigned to the following working groups: (1) the control group (*n* = 10); the control group was fed a ND, (2) HFD group (*n* = 10), (3) HFD + 50 mg/kg rosmarinic acid group (*n* = 10), (4) HFD + 100 mg/kg rosmarinic acid group (*n* = 10), and (5) HFD + 100 mg/kg metformin group (*n* = 10). Rosmarinic acid (>98% purified by high-performance liquid chromatography) and metformin hydrochloride were dissolved in distilled water and administered via gastric gavage once a day for 8 weeks. Once treatment was complete, the mice were fasted overnight, FBG levels were measured, and an oral glucose tolerance test (OGTT) was performed. The mice were then sacrificed, blood samples were collected by heart puncture with heparinized syringes and centrifuged at 3000× *g* for 20 min, and the supernatants were stored at −80 °C for biochemical analysis. Liver tissues (30 mg) were homogenized in 200 µL of NP40 substitute assay reagent (Cayman Chemicals) containing protease inhibitors (Thermo Fisher Scientific, Waltham, MA, USA) and centrifuged for 10 min at 10,000× *g* at 4 °C to collect the supernatants for hepatic cholesterol and triglyceride assays.

### 2.3. Blood Glucose Determination

Fasting blood glucose levels were determined from tail vein blood samples using an Accu-Check Glucometer (Roche Diagnostics, Mannheim, Germany) following 12 h of mouse fasting. In the oral glucose tolerance test (OGTT) experiment, mice were fasted for 12 h and were given 2 mg/kg body weight oral glucose administration. After 2 h, the glucose levels were measured in samples from tail vein blood.

### 2.4. Plasma Total Cholesterol and Triglyceride Quantification

At the sacrifice of mice, blood samples were collected by heart puncture, and plasma total cholesterol and triglycerides were determined in plasma by using a cholesterol fluorometric assay kit and triglyceride colorimetric assay kit, respectively, according to the manufacturer’s instructions. Briefly, blood samples collected with heparinized syringes were centrifuged at 3000× *g* for 20 min at 4 °C. The supernatants were carefully transferred to e-tubes and stored at −80 °C until the analysis of total cholesterol and triglycerides. The plasma sample and enzyme cocktail were mixed and incubated for 30 min at 37 °C. The triglyceride was measured by detecting the absorbance at 540 nm, while the level of cholesterol was determined by reading the fluorescence at 540 nm of excitation and 590 nm of emission using a VersaMax microplate reader (Molecular Devices, San Jose, CA, USA).

### 2.5. Hepatic Cholesterol and Triglyceride Assay

Hepatic cholesterol and triglyceride contents were measured using a Cholesterol Fluorometric Assay Kit and Triglyceride Colorimetric Assay Kit, respectively, according to the manufacturer’s instructions. Briefly, liver homogenates were centrifuged for 10 min at 10,000× *g* at 4 °C as described in Section 2.2, and the supernatants were collected for the assay. The sample and enzyme mixture were incubated for 30 min at 37 °C, and the absorbance was measured at 540 nm using a VersaMax microplate reader (Molecular Devices, San Jose, CA, USA) for triglycerides. Fluorescence was read at 540 nm excitation and 590 nm emission for cholesterol.

### 2.6. Hematoxylin and Eosin Staining

The liver tissues were fixed in 10% formalin solution, embedded in paraffin and sectioned at 5 µm thickness. The sections were stained with hematoxylin and eosin (H&E) according to the standard protocols for histological analysis, and all staining images were captured using a CKX41 light microscope (Olympus, Tokyo, Japan).

### 2.7. Immunoblotting

Liver tissues were homogenized in ice-cold RIPA buffer with protease inhibitors (Thermo Fisher Scientific), sonicated, and incubated for 20 min on ice. After centrifugation, the supernatant was transferred to a clean tube, and the protein concentration was determined using a Pierce^TM^ BCA protein assay kit (Thermo Fisher Scientific). The protein lysates were separated using sodium dodecyl sulfat-polyacrylamide gel electrophoresis, transferred to polyvinylidene difluoride membranes, and blocked with 5% skim milk or 3% bovine serum albumin. The membranes were incubated with primary antibodies in blocking buffer solution at 4 °C overnight. Next, the membranes were incubated with the appropriate horseradish peroxidase-conjugated secondary antibodies (Bio-Rad, Hercules, CA, USA) at room temperature for 1 h and then visualized with ECL substrates (Bio-Rad). The ChemiDoc^TM^ XRS+ System (Bio-Rad) was used to evaluate the density of protein bands, and relative protein levels were quantified using Image Lab^TM^ software (Bio-Rad). The relative protein levels are presented after normalization to actin as a loading control.

### 2.8. Data and Statistical Analysis

All data were analyzed by using GraphPad Prism 7 software (GraphPad Software, San Diego, CA, USA). Statistical comparisons were made by using one-way ANOVA followed by Tukey’s multiple comparisons test for comparing three or more groups. All results are presented as the means ± standard error of the mean (SEM) and were considered statistically significant if *p* < 0.05.

## 3. Results

### 3.1. Rosmarinic Acid Reduces Body Weight and Blood Glucose in HFD-Fed Mice

Some researchers have studied the effect of rosmarinic acid in animal model at various dose ranges between 2.5 and 200 mg/kg [23,27,28,29,30]. Our previous in vitro study showed that rosmarinic acid effectively reduced oxLDL-induced cholesterol contents under HG conditions in macrophages at 50 and 100 μM [26]. Based on these reports, we investigated the protective effects of rosmarinic acid on hyperlipidemia in HFD-fed C57BL/6 mice at 50 mg/kg and 100 mg/kg. In addition, metformin is the first-line medication for the treatment of type 2 diabetes, particularly in people who are overweight [31,32]. Furthermore, it was recently reported that metformin possesses a lipids-lowering effect [33]. Therefore, metformin was used as a control for comparison. C57BL/6 mice were fed a ND or HFD for 8 weeks to induce diabetic conditions and then treated with or without rosmarinic acid (50 mg/kg/day or 100 mg/kg/day) or metformin (100 mg/kg/day) for 8 more weeks (Figure 1A). After 16 weeks, HFD feeding significantly increased the body weight. In contrast, rosmarinic acid treatment (100 mg/kg/day) reduced the body weight gain observed in HFD-fed mice (Figure 1B), which was small but a significant change. This change in body weight was not due to decreased food intake as there was no significant change in food intake between HFD-fed mice and HFD-fed mice treated with rosmarinic acid or metformin (data not shown). In addition, hyperglycemia, which was assessed based on the FBG levels and OGTT results of HFD-fed mice, was alleviated by 50 mg/kg/day and 100 mg/kg/day rosmarinic acid (Figure 1C,D). Interestingly, the effect of rosmarinic acid on body weight and blood glucose was as efficient as that of metformin (Figure 1B–D).

### 3.2. Rosmarinic Acid Inhibits Plasma Lipid and Hepatic Lipid Accumulation in HFD-Fed Mice

Next, we investigated the effect of rosmarinic acid on lipid profiles such as plasma and liver lipids in HFD-fed mice. HFD-fed mice showed higher levels of plasma total cholesterol (Figure 2A) and triglycerides (Figure 2B) than did ND-fed mice, which were significantly reduced by rosmarinic acid treatment at doses of 50 mg/kg per day and 100 mg/kg per day. Similarly, rosmarinic acid administration reduced hepatic cholesterol (Figure 2C) and triglycerides (Figure 2D), which were induced in HFD-fed mice. Then, we analyzed hepatic lipid accumulation by H&E. We observed hepatic lipid accumulation in liver tissues from HFD-fed mice, which was significantly reduced by rosmarinic acid treatment (Figure 2E,F). These findings suggest that rosmarinic acid exhibits a protective effect against hyperlipidemia and hyperglycemia in HFD-fed mice.

### 3.3. Rosmarinic Acid Increases Hepatic LDL-R and SR-BI Protein Expression

Then, we tried to elucidate the possible mechanisms by which rosmarinic acid reduces HFD-induced hyperlipidemia. The expression of liver low-density lipoprotein receptor (LDL-R) and HDL receptor SR-BI has an indispensable role in the clearance of lipoproteins from blood [12,13]. SR-BI and LDL-R facilitate hepatic uptake of HDL cholesterol and non-HDL cholesterol, respectively, constituting a major component of the RCT pathway. Western blot analysis revealed that HFD-fed mouse liver tissue showed a significantly lower expression of SR-BI protein, whereas LDL-R protein expression was slightly increased. In contrast, rosmarinic acid treatment markedly augmented hepatic SR-BI and LDL-R protein expression (Figure 3A–C; Appendix A). These results suggest that rosmarinic acid may facilitate plasma cholesterol clearance by increasing its uptake by the liver.

### 3.4. Rosmarinic Acid Enhances Cholesterol Excretion by Regulating ABCA1 and ABCG5/8 in the Mouse Liver

To examine whether the reductive effect of rosmarinic acid on hepatic lipid accumulation was associated with enhanced cholesterol excretion, we determined the effect of rosmarinic acid on the expression of ABCG5 and ABCG8 proteins, key transporters involved in hepatobilial cholesterol excretion, in the mouse liver. The results showed that rosmarinic acid treatment significantly increased ABCA1, ABCG5, and ABCG8 protein expression in the liver at doses of 50 mg/kg and 100 mg/kg, respectively (Figure 4A–D; Appendix A). Then, we investigated whether rosmarinic acid treatment affects CYP7A1 protein expression because CYP7A1 protein is a key rate-limiting enzyme for bile acid biosynthesis in the liver that is responsible for the final elimination of cholesterol by the liver. Figure 4E,F shows that rosmarinic acid significantly increased the level of CYP7A1 protein in the mouse liver tissue, suggesting that rosmarinic acid promotes cholesterol excretion by the liver (Appendix A).

### 3.5. Rosmarinic Acid Increases Fatty Acid β-Oxidation in HFD-Fed Mice

Next, we investigated whether rosmarinic acid reduces hepatic liver accumulation by modulating lipid metabolism. AMPK activation is a key regulator of the fatty acid β-oxidation process that is involved in lipid metabolism [34]. The effect of rosmarinic acid on the phosphorylation of AMPK (Thr172) was therefore determined. The levels of phospho-AMPK were significantly decreased in HFD-fed mice compared to normal diet-fed mice, whereas HFD-fed mice treated with rosmarinic acid exhibited increased phospho-AMPK levels (Figure 5A,B; Appendix A), entailing the activation of the AMPK signaling pathway. Furthermore, AMPK has been shown to enhance lipid metabolism by regulating carnitine palmitoyl transferase 1 (CPT1) A activation for fat oxidation [21]. Our study also showed that CTP1A protein expression was not changed in HFD-fed mice, but rosmarinic acid treatment significantly augmented CPT1A protein levels in HFD-fed mice (Figure 5C,D; Appendix A). These results suggest that rosmarinic acid activates the AMPK/CPT1A pathway to facilitate fatty acid β-oxidation in HFD-fed mice.

## 4. Discussion

Hyperlipidemia is a well-established strong risk factor for atherosclerosis and cardiovascular diseases. Lipid-modifying drugs, which primarily lower plasma levels of LDL cholesterol, substantially reduce the risk of cardiovascular diseases [35,36]. Drugs from natural products have distinct advantages, such as extensive resources and fewer adverse effects as well as the lipids-lowering action [37,38]. Our study found that rosmarinic acid improved abnormal serum (Figure 2A,B) and hepatic lipid levels (Figure 2C,D) in HFD-induced dyslipidemia in mice, and the effects were similar to or better than those of metformin. Based on the study findings, we sought to determine the molecular mechanism pertaining to the lipid-lowering effect of rosmarinic acid and found that the protective effect of rosmarinic acid is related to the promotion of RCT. Cholesterol and related lipid homeostasis is a complex network involving biosynthesis, uptake, transport, and esterification that is tightly controlled to avoid excess cholesterol accumulation and eventually metabolic disease establishment [4,7]. The liver is the only organ that can excrete significant amounts of cholesterol, either by converting it into bile salts or by secreting it as unesterified cholesterol into bile and subsequently excreting via feces [5,39]. Hepatic uptake of cholesterol from the circulation is a critical step of the RCT pathway [40], an indispensable mechanism involved in mitigating hyperlipidemia [10] and cardiovascular diseases [41]. Circulating cholesterol transported by LDL is taken from the circulation through liver LDL-R [13]. Alternatively, cholesterol effluxed from peripheral cells is incorporated into HDL molecules and delivered to the liver through the scavenger receptor SR-BI protein [12]. Human studies and experiments using transgenic and gene knockout mice have demonstrated that increased expression of LDL-R and SR-BI protects against hyperlipidemia and diverse metabolic disorders [12,42,43,44,45]. In concordance, we found that rosmarinic acid treatment significantly increased SR-BI protein expression, which was downregulated in HFD-fed mice. Furthermore, rosmarinic acid enhanced LDL-R protein expression in HFD-fed mice, suggesting that it increased the uptake of circulating cholesterol by the liver (Figure 3). It is reported that plasma cholesterol is mainly transported by HDL particles in mice, unlike in humans, where LDL particles transport plasma cholesterol [46]. Therefore, there may be limitations to using a mouse model to study lipoprotein metabolism and make inferences about the human condition. Although there are some minor protein-based metabolic differences between mouse and human species, mouse and human lipoproteome studies have shown that the protein diversity in the LDL and HDL size ranges, major HDL proteins, and apolipoprotein A-I was similar in mice and humans [47]. Moreover, von Scheidt et al. [48] showed that genes involved in lipids metabolism and transport overlap between mouse and human species. Therefore, mouse model is used to investigate lipoprotein metabolism, which can serve as the basis for making some decisions about human conditions.

We observed that HFD-fed mice had elevated levels of cholesterol and triglycerides in liver tissues. Rosmarinic acid treatment significantly diminished liver lipid levels and accumulation in HFD-fed mice (Figure 2C–F). Therefore, we wondered whether the protective effect of rosmarinic acid was due to increased cholesterol metabolism and excretion from the liver into bile. The biliary secretion of cholesterol is crucial for the maintenance of cholesterol homeostasis and constitutes a major defense against the accumulation of cholesterol in blood and tissues. Liver proteins ABCG5 and ABCG8 were reported to form obligate heterodimers and play indispensable roles in mediating liver cholesterol excretion in humans and mice [14,49,50,51]. Molusky et al. [51] reported that the cardioprotective effect of metformin was due to upregulation of hepatic ABCG5/8 transporters and LDL-R and subsequent reverse cholesterol transport in C57BL/6J mice fed a western diet. Consistently, the results of this study show that rosmarinic acid induced the expression of hepatic ABCG5 and ABCG8 proteins in HFD-fed mice, suggesting that rosmarinic acid facilitates hepatobiliary cholesterol excretion. Interestingly, the effect of rosmarinic acid on the expression of hepatic ABCG5 and ABCG8 proteins was superior to the effect of metformin (Figure 4A–D). In addition, we observed that rosmarinic acid treatment significantly increased ABCA1 expression in the liver tissues of HFD-fed mice, even though ABCA1 protein expression revels remained unchanged in HFD-fed mice compared to normal diet-fed mice. ABCA1, predominantly expressed in the liver [52,53], is important to maintain hepatic cholesterol homeostasis by promoting hepatic cholesterol efflux [54]. Notably, liver ABCA1 is vital for the synthesis of apoA-I, the formation of nascent HDL, and the generation of mature HDL [55,56], which are all involved in the initiation of the RCT pathway [57]. Moreover, overexpression of human ABCA1 in the liver and macrophages of C57Bl/6 mice reduced atherosclerosis [58]. Thus, these findings confirm that rosmarinic acid modulates the liver cholesterol homeostasis process. Furthermore, rosmarinic acid stimulated the induction of CYP7A1, the rate-limiting enzyme of cholesterol metabolism and bile acid synthesis (Figure 4E,F). The overexpression of CYP7A1 maintained cholesterol homeostasis through hepatic bile acid synthesis and secretion [16]. Transgenic mice overexpressing CYP7A1 were protected from atherosclerosis [59] and liver inflammation and fibrosis [60]. Moreover, the hypocholesterolemic effect of curcumin was attributed to CYP7A1 upregulation [61]. Correspondingly, this study suggests that the hypolipidemic effect of rosmarinic acid is partly due to its ability to induce hepatic CYP7A1 expression.

AMPK is increasingly recognized as a potent inhibitor of hyperlipidemia, mainly through modulation of lipid metabolism by the liver. Dusabimana et al. [21] reported that hepatic AMPK activation induced fatty acid oxidation, prevented lipid accumulation in HFD-fed mice, and ameliorated nonalcoholic fatty liver disease (NAFLD). AMPK activation inhibits lipid accumulation and de novo lipid synthesis by inactivating acetyl-CoA carboxylase (ACC) and stimulates fatty acid oxidation by upregulating CPT1 [20,62]. A previous report showed that rosmarinic acid prevented NAFLD by maintaining mitochondrial integrity [63]. Vasileva et al. [64] claimed that rosmarinic acid inhibited adipogenic genes and proteins in human adipocytes. Our results showed that rosmarinic acid treatment activated AMPK and increased the expression of CPT1A protein, which are the key inducers of fatty acid β-oxidation and subsequent reduction of lipid levels in the liver of HFD-fed mice.

The present study has some limitations that should be addressed in future experiments. The present study focused only on lipid trafficking in vivo. The study did not examine the effect of rosmarinic acid on ABCG5 and ABCG8 expression levels in the intestine to delineate their involvement in limiting intestinal cholesterol absorption and the final step of the RCT pathway, and further study is needed.

## 5. Conclusions

This study revealed that rosmarinic acid possesses lipid-lowering effects. Rosmarinic acid improved the lipid profile in HFD-fed mice, promoted RCT, and mediated fatty acid oxidation. Thus, rosmarinic acid could protect against hyperlipidemia and associated cardiovascular diseases. Our data emphasize the potential therapeutic implications of rosmarinic acid against dyslipidemia and derived metabolic disorders using an in vivo mouse model (Figure 6).

## Figures and Tables

**Figure 1 biomolecules-11-01470-f001:**
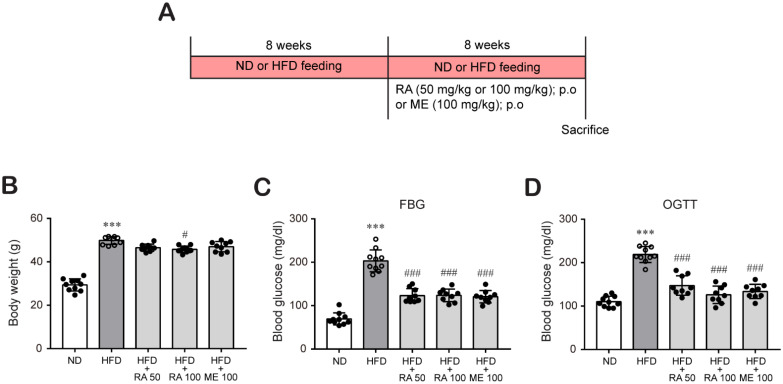
RA reduces body weight and blood glucose levels in HFD-fed mice. C57BL/6 mice were fed an ND or HFD (60% kcal fat) for 8 weeks to induce diabetic conditions as described in the Methods section. Then, the mice were randomly assigned to the following groups: (1) the control group (ND, *n* = 10), (2) HFD group (*n* = 10), (3) HFD + 50 mg/kg rosmarinic acid group (*n* = 10), (4) HFD + 100 mg/kg rosmarinic acid group (*n* = 10), and (5) HFD + 100 mg/kg metformin group (*n* = 10). Rosmarinic acid and metformin were administered orally once a day for 8 weeks. (**A**) Experimental scheme to induce obesity and diabetes in C57BL/6 mice. (**B**) Body weights were measured twice a week, and final body weights before sacrifice are presented as the mean ± SD. (**C**,**D**) FGB levels (**C**) and OGTT values (**D**) after oral gavage of d-glucose (2 g/kg) were measured in blood samples obtained from the tail vein. The results are expressed as the mean ± SD. *** *p* < 0.001 vs. ND control mice; # *p* < 0.05, ### *p* < 0.001 vs. HFD mice.

**Figure 2 biomolecules-11-01470-f002:**
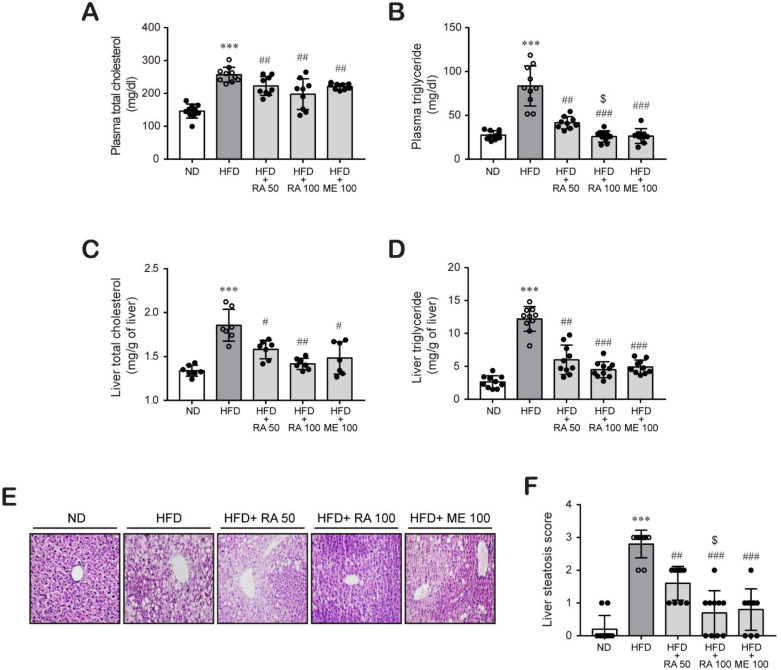
RA reduces plasma lipid and hepatic lipid accumulation and cellular injury in HFD-fed mice. Plasma total cholesterol (**A**) and triglyceride (**B**) levels were quantified in blood samples collected by heart puncture at sacrifice (*n* = 9–10). (**C**) Hepatic total cholesterol (*n* = 7) and (**D**) triglyceride levels were determined in liver tissue (*n* = 9–10) as described in the Methods section. Data are presented as the mean ± SD. *** *p* < 0.001 vs. ND control mice; # *p* < 0.05, ## *p* < 0.01, ### *p* < 0.001 vs. HFD mice; $ *p* < 0.05 vs. RA 50. (**E**,**F**) H&E staining was performed to assess lipid accumulation in liver tissues (*n* = 10) (**E**), and histological scores for steatosis ranged from 0 to 3 in whole-value increments, where 0 = 0–5%, 1 = 6–33%, 2 = 34–66%, and 3 = 67–100% of hepatocytes positive for steatosis (**F**). Data are presented as the mean ± SEM. *** *p* < 0.001 vs. ND control mice; ## *p* < 0.01, ### *p* < 0.001 vs. HFD mice; $ *p* < 0.05 vs. RA 50. Scale bar, 100 µm.

**Figure 3 biomolecules-11-01470-f003:**
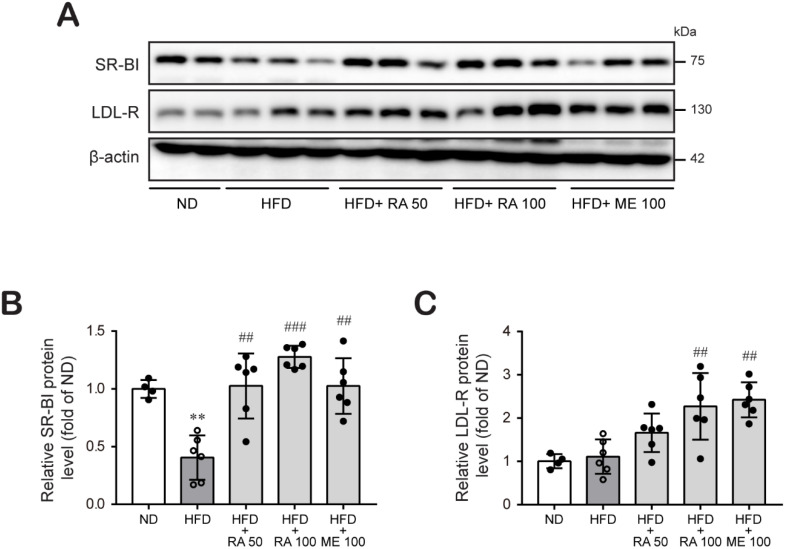
RA induces hepatic LDL-R and SR-BI protein expression in HFD-fed mice. (**A**–**C**) Liver tissues were lysed, and the levels of proteins regulating hepatic lipid uptake, LDL-R (**A**,**B**) and SR-BI (**A**,**C**), were examined by western blot analysis. β-actin was used as a loading control. Band densities were quantified, and the relative protein levels are presented as the mean ± SD (*n* = 4–6). ** *p* < 0.01 vs. ND control mice; ## *p* < 0.01, ### *p* < 0.001 vs. HFD mice.

**Figure 4 biomolecules-11-01470-f004:**
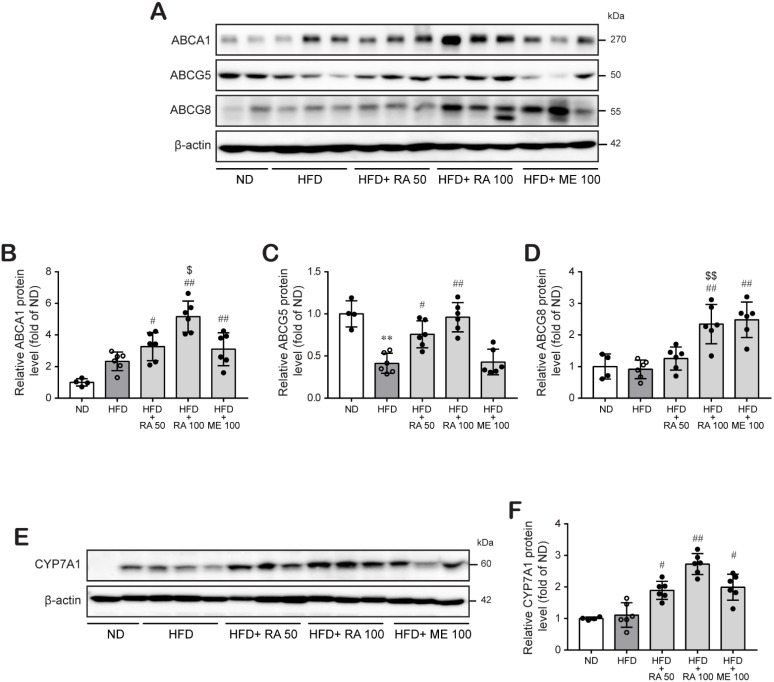
RA significantly induces ABCG5/8 and ABCA1 protein levels, which are involved in hepatobilial cholesterol excretion, and CYP7A1 protein levels, a key rate-limiting enzyme for bile acid biosynthesis in HFD-fed mice. (**A**–**D**) ABCG5, ABCG8, ABCA1, and (**E**,**F**) CYP7A1 protein levels were examined in liver tissues by western blot analysis, and the relative protein levels are presented as the mean ± SD (*n* = 4–6). ** *p* < 0.01 vs. ND control mice; # *p* < 0.05, ## *p* < 0.01, vs. HFD mice; $ *p* < 0.05, $$ *p* < 0.01 vs. RA 50.

**Figure 5 biomolecules-11-01470-f005:**
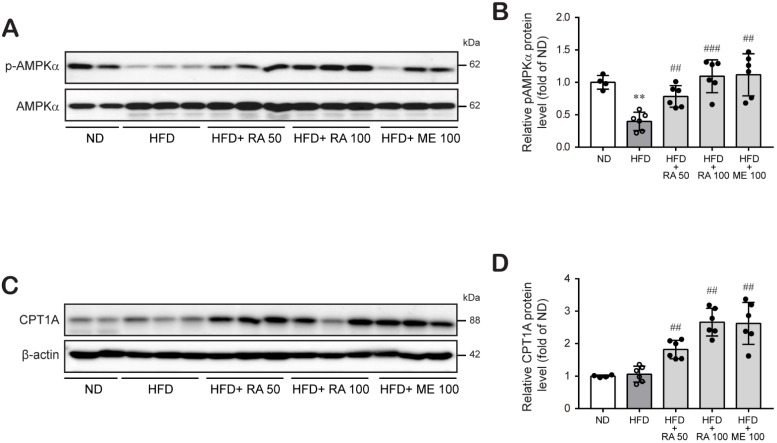
RA enhances fatty acid β-oxidation by activating the AMPK-CPT1A pathway in HFD-fed mice. (**A**–**D**) Liver tissues were lysed to perform western blot analysis to determine the protein levels of phospho-AMPKα, AMPKα (**A**,**B**), and CPT1A (**C**,**D**). Data are presented as the mean ± SD (*n* = 4–6) (**B**). ** *p* < 0.01 vs. ND control mice; ## *p* < 0.01, ### *p* < 0.001 vs. HFD mice vs. RA 50.

**Figure 6 biomolecules-11-01470-f006:**
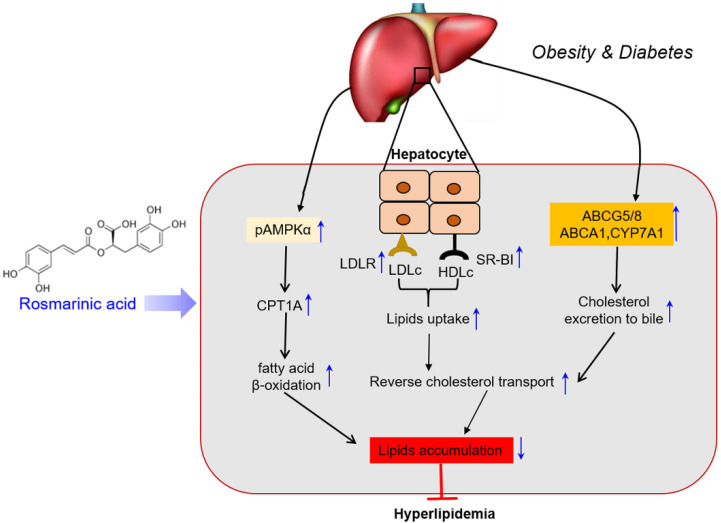
Schematic presentation by which RA induces cholesterol excretion and protects the liver in HFD-fed mice. A schematic diagram illustrating the protective mechanisms of rosmarinic acid against hyperlipidemia. Rosmarinic acid enhances reverse cholesterol transport (RCT) and lipid metabolism. Rosmarinic acid mediates RCT through increased plasma cholesterol uptake and subsequent hepatobiliary excretion and induces lipid metabolism by increasing fatty acid β-oxidation in AMPK-dependent activation of CPT1A, thus modulating lipid accumulation and attenuating hyperlipidemia in HFD-fed mice.

## Data Availability

The data that support the findings of this study are available from the corresponding author upon reasonable request.

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
