# Peer review of "Rosmarinic Acid Exhibits a Lipid-Lowering Effect by Modulating the Expression of Reverse Cholesterol Transporters and Lipid Metabolism in High-Fat Diet-Fed Mice"

_biomolecules, 2021, doi:10.3390/biom11101470_

Round 1

Reviewer 1 Report

The article from Nyandwi et al. investigated the effect of a natural drug rosmarinic acid in high-fat diet-fed mice to treat hyperlipidemia. It is impressive that rosmarinic acid at the concentration tested reduced lipid accumulation in plasma and liver in HDL-fed mice. The authors showed that rosmarinic acid modulates the reverse cholesterol transport process that it increased the expression of cholesterol uptake-related receptors in the liver including SR-B1 and LDL-R. It also increased the expression of cholesterol excretion transporters ABCG5, ABCG8, and cholesterol metabolism enzyme CYP7A1. Furthermore, it activates fatty acid oxidation by increasing AMPA and CTP1A. Based on this study the lipid-reducing effect of rosmarinic acid and the mechanism through which it acts are promising and interesting. The experiments are reasonably designed, the results were analyzed/presented with sufficient details, and the manuscript is written in a logical way. One minor command:

Line 193-194: “…then treated with or without rosmarinic acid (50 mg/kg/day or 100 mg/kg/day) or metformin (100 mg/kg/day) for 8 more weeks…”  It is better to consider discuss/explain why the above concentrations are selected in the study. Also, explain and add reference why metformin is chosen as one of the control here.

Author Response

Reviewer 1

The article from Nyandwi et al. investigated the effect of a natural drug rosmarinic acid in high-fat diet-fed mice to treat hyperlipidemia. It is impressive that rosmarinic acid at the concentration tested reduced lipid accumulation in plasma and liver in HDL-fed mice. The authors showed that rosmarinic acid modulates the reverse cholesterol transport process that it increased the expression of cholesterol uptake-related receptors in the liver including SR-B1 and LDL-R. It also increased the expression of cholesterol excretion transporters ABCG5, ABCG8, and cholesterol metabolism enzyme CYP7A1. Furthermore, it activates fatty acid oxidation by increasing AMPA and CTP1A. Based on this study the lipid-reducing effect of rosmarinic acid and the mechanism through which it acts are promising and interesting. The experiments are reasonably designed, the results were analyzed/presented with sufficient details, and the manuscript is written in a logical way. One minor command:

Line 193-194: “…then treated with or without rosmarinic acid (50 mg/kg/day or 100 mg/kg/day) or metformin (100 mg/kg/day) for 8 more weeks…” It is better to consider discuss/explain why the above concentrations are selected in the study. Also, explain and add reference why metformin is chosen as one of the control here.

-> Answer: Thank you for your valuable comment. We added why we chose the dose of RA (50 mg/kg/day or 100 mg/kg/day) and metformin as a control for comparison as follows; “Some researchers have studied the effect of rosmarinic acid in animal model at various dose ranges between 2.5 ~ 200 mg/kg [27-31]. Our previous in vitro study showed that rosmarinic acid effectively reduced oxLDL-induced cholesterol contents under HG conditions in macrophages at 50 and 100 mM [26]. Based on these reports, we investigated the protective effects of rosmarinic acid on hyperlipidemia in HFD-fed C57BL/6 mice at 50 mg/kg and 100 mg/kg. In addition, metformin is the first-line medication for the treatment of type 2 diabetes, particularly in people who are overweight [32,33]. Furthermore, it was recently shown that metformin possesses lipids lowering effect [34]. Therefore, metformin was used as a control for comparison.” (Please see lines 191~199 and references).

<References>

  1. Baba, S.; Osakabe, N.; Natsume, M.; Terao, J. Orally administered rosmarinic acid is present as the conjugated and/or methylated forms in plasma, and is degraded and metabolized to conjugated forms of caffeic acid, ferulic acid and m-coumaric acid. Life Sci. 2004, 75, 165-178.
  2. Karthik, D.; Viswanathan, P.; Anuradha, C.V. Administration of rosmarinic acid reduces cardiopathology and blood pressure through inhibition of p22phox NADPH oxidase in fructose-fed hypertensive rats. J. Cardiovasc. Pharmacol. 2011, 58:514-521.
  3. Li, G.S.; Jiang, W.L.; Tian, J.W.; Qu, G.W.; Zhu, H.B.; Fu, F.H. In vitro and in vivo antifibrotic effects of rosmarinic acid on experimental liver fibrosis. Phytomedicine 2010, 17, 282-288.
  4. Nakazawa, T.; Ohsawa, K. Metabolites of orally administered Perilla frutescens extract in rats and humans. Biol. Pharm. Bull. 2000, 23, 122-127.
  5. Osakabe, N.; Yasuda, A.; Natsume, M.; Sanbongi, C.; Kato, Y.; Osawa, T.; Yoshikawa, T. Rosmarinic acid, a major polyphenolic component of Perilla frutescens, reduces lipopolysaccharide (LPS)-induced liver injury in D-galactosamine (D-GalN)-sensitized mice. Free Radic. Biol. Med. 2002, 33, 798-806.
  6. Maruthur, N.M.; Tseng, E.; Hutfless, S.; Wilson, L.M.; Suarez-Cuervo, C.; Berger, Z.; Chu, Y.; Iyoha, E.; Segal, J.B.; Bolen, S. Diabetes Medications as Monotherapy or Metformin-Based Combination Therapy for Type 2 Diabetes: A Systematic Review and Meta-analysis. Ann. Intern. Med. 2016, 164, 740-751.
  7. Kopelman, P.G.; Caterson, I.D.; Dietz, W.H. Clinical Obesity in adults and choldren, 2nd ed, Oxford: John Wiley & Sons Ltd. 2008, p262.
  8. Hu, D.; Guo, Y.; Wu, R.; Shao, T.; Long, J.; Yu, B.; Wang, H.; Luo, Y.; Lu, H.; Zhang, J.; Chen, Y.E.; Peng, D. New Insight into Metformin-Induced Cholesterol-Lowering Effect Crosstalk between Glucose and Cholesterol Homeostasis via ChREBP (Carbohydrate-Responsive Element-Binding Protein)-Mediated PCSK9 (Proprotein Convertase Subtilisin/Kexin Type 9) Regulation. Arterioscler. Thromb. Vasc. Biol. 2021, 41, e208-e223.

Reviewer 2 Report

The manuscript is of interest because it investigates the possibility that a natural compound might ameliorate lipid profile in high fat-fed mice, primarily by looking at reverse cholesterol transport, an important issue in cholesterol metabolism.

Here are my suggestions.

References are sometimes inappropriate or obsolete (i.e. ref. 1,2,6,7 in the Introduction). Also References in other parts of the manuscript should be checked. For example, references 32,33 (line 286) and 4,34 (line 287) are far from referring to major milestone articles in the field of the lipid theory of cardiovascular disease.

It is not clear whether HFD mice were 37, 45 or 40 as reported in Materials and Methods and legend to Figure 1.

Body weight reduction in the treatment groups, although statistically significant, seems rather small (maybe < 10%, according to figure 1B). The authors should clearly state that it is a small weight reduction. The authors should also disclose whether weight reduction was not to be ascribed to a reduced food intake in the treatment groups.

In the Discussion, the sentence reported in lines 294 and 295 is highly controversial and weakly supported by scientific evidences in humans. Are the authors meaning that natural products are better drugs than non-natural products, such as statins? I would be more careful in conveying such a message.

The authors should make a comment about the usefulness of the metformin-treated experimental group.

The authors should make a comment about using the mouse model to study lipoprotein metabolism and make inference about human conditions, since mice are carrying plasma cholesterol mainly in HDL particles compared to humans where LDL are the major plasma cholesterol carriers. Results could have been more convincing if also HDL and LDL-cholesterol had been measured.

English should be improved, and here follow some examples:

Line 16-impair hyperlipidemia

Line 46-Lipids …… are a complex process

Line 53-RCTs play a detrimental role

Line 353- … impaired nonalcoholic fatty liver disease

Author Response

Reviewer 2

The manuscript is of interest because it investigates the possibility that a natural compound might ameliorate lipid profile in high fat-fed mice, primarily by looking at reverse cholesterol transport, an important issue in cholesterol metabolism.

Here are my suggestions.

References are sometimes inappropriate or obsolete (i.e. ref. 1,2,6,7 in the Introduction). Also References in other parts of the manuscript should be checked. For example, references 32,33 (line 286) and 4,34 (line 287) are far from referring to major milestone articles in the field of the lipid theory of cardiovascular disease.

-> Answer: Thank you for your comments. We revised our manuscript again and agreed with your comment that those references (1, 2, 6, 7 in the Introduction) are not needed here. So, we deleted those references. In addition, we also deleted references 2, 4, 32, 33, 34 in the Discussion and modified that paragraph (please lines 295~297).

It is not clear whether HFD mice were 37, 45 or 40 as reported in Materials and Methods and legend to Figure 1.

-> Answer: Thank you for your indication. There was an error. The number which was described in the Figure 1 legend is right. During revision, we corrected the number in Materials and Methods (please see lines 112, 118).

Body weight reduction in the treatment groups, although statistically significant, seems rather small (maybe < 10%, according to figure 1B). The authors should clearly state that it is a small weight reduction. The authors should also disclose whether weight reduction was not to be ascribed to a reduced food intake in the treatment groups.

-> Answer: As you suggested, we added that statement as follows; “In contrast, rosmarinic acid treatment (100 mg/kg/day) reduced the body weight gain observed in HFD-fed mice (Figure 1B), which was small but a significant change. This change in body weight was not due to decreased food intake as there was no significant change in food intake between HFD-fed mice and HFD-fed mice treated with rosmarinic acid or metformin (data not shown). Please see lines 202~207.

In the Discussion, the sentence reported in lines 294 and 295 is highly controversial and weakly supported by scientific evidences in humans. Are the authors meaning that natural products are better drugs than non-natural products, such as statins? I would be more careful in conveying such a message.

-> Answer: We fully understand your concern. In fact, we intended to highlight the advantages of natural product. Natural products such as rosmarinic acid would be easily accessible; they are found in commonly edible plants and their safety is presumably granted. In addition, natural products are traditionally used to manage different metabolic disorders including hyperlipidemia. We don’t think that natural products are superior to non-natural products in terms of efficacy or potency because those parameters were not investigated. Therefore, according to your comment, we deleted the sentence and modified that paragraph in the Discussion. Please see the 1st paragraph in the Discussion. Please see lines 297~302. Thank you for your comment.

The authors should make a comment about the usefulness of the metformin-treated experimental group.

-> Answer: According to your suggestion, we described the reason why we used metformin as a positive control in the lines 196~199 as follows, “In addition, metformin is the first-line medication for the treatment of type 2 diabetes, particularly in people who are overweight [32,33]. Furthermore, it was recently reported that metformin possesses lipids lowering effect [34]. Therefore, metformin was used as a control for comparison.”

In addition, we compared the effect of rosmarinic acid on body weight and blood glucose with those of metformin. Please see lines 209, 210, “Interestingly, the effect of rosmarinic acid on body weight and blood glucose was as efficient as metformin (Figure 1B~D).”

Furthermore, we also inserted some sentences comparing the effects between rosmarinic acid and metformin as follows, “Our study found that rosmarinic acid improved abnormal serum (Figure 2A, B) and hepatic lipid levels (Figure 2C, D) in HFD-induced dyslipidemia in mice, which effects were as similar as or better than those of metformin.” (lines 299~301) and “Molusky et al. [51] reported that the cardioprotective effect of metformin was due to upregulation of hepatic ABCG5/8 transporters and LDL-R and subsequent reverse cholesterol transport in C57BL/6J mice fed a western diet. Consistently, the results of this study show that rosmarinic acid induced the expression of hepatic ABCG5 and ABCG8 proteins in HFD-fed mice, suggesting that rosmarinic acid facilitates hepatobiliary cholesterol excretion. Interestingly, the effect of rosmarinic acid on the expression of hepatic ABCG5 and ABCG8 proteins was superior to the effect of metformin (Figure 4A~D).” (lines 340~347)

The authors should make a comment about using the mouse model to study lipoprotein metabolism and make inference about human conditions, since mice are carrying plasma cholesterol mainly in HDL particles compared to humans where LDL are the major plasma cholesterol carriers. Results could have been more convincing if also HDL and LDL-cholesterol had been measured.

-> Answer: Thank you for your valuable comments.

According to your suggestion, we inserted those comments in the Discussion part as follows; “It is reported that plasma cholesterol is mainly transported by HDL particles in mice, unlike humans, where LDL particles transport plasma cholesterol [47]. Therefore, there may be limitations to use a mouse model to study lipoprotein metabolism and make inferences about the human condition. Although there are some minor protein-based metabolic differences between mouse and human species, mouse and human lipoproteome studies have shown that the protein diversity in the LDL and HDL size ranges, major HDL proteins, and apolipoprotein A-I was similar in mice and humans [48]. Moreover, von Scheidt et al. [49] showed that genes involved in lipids metabolism and transport overlap between mouse and human species. Therefore, mouse model is used to investigate lipoprotein metabolism, which can serve as the basis for making some decisions about human conditions.” (please see lines 321~331).

<References>

  1. Gordon, S.M.; Li, H.; Zhu, X.; Shah, A.S.; Lu, L.J.; Davidson, W.S. A Comparison of the mouse and human lipoproteome: suitability of the mouse model for studies of human lipoproteins. J. Proteome Res. 2015, 14, 2686-2695.
  2. Karimi, I. Animal Models as Tools for Translational Research: Focus on Atherosclerosis, Metabolic Syndrome and Type-II Diabetes Mellitus. In book: Lipoproteins: Role in Health and Diseases. Intech. 2012. DOI:10.5772/47769
  3. von Scheidt, M.; Zhao, Y.; Kurt, Z.; Pan, C.; Zeng, L.; Yang, X.; Schunkert, H.; Lusis, A.J. Applications and Limitations of Mouse Models for Understanding Human Atherosclerosis. Cell Metab. 2017, 25, 248–261.

English should be improved, and here follow some examples:

Line 16-impair hyperlipidemia

Line 46-Lipids …… are a complex process

Line 53-RCTs play a detrimental role

Line 353- … impaired nonalcoholic fatty liver disease

-> Answer: Thank you for your kindness. We reviewed your manuscript and tried to improve English including what you indicated (please see the locations; lines 15; 45; 52; 367).
